# Nutritional Strategies for the Treatment and Prevention of Sepsis Outside the Intensive Care Unit

**DOI:** 10.3390/nu16233985

**Published:** 2024-11-21

**Authors:** Maurizio Gabrielli, Raffaella Zaccaria, Michele Impagnatiello, Lorenzo Zileri Dal Verme, Antonio Gasbarrini

**Affiliations:** Department of Medical and Surgical Sciences, Fondazione Policlinico Universitario A. Gemelli IRCCS, Università Cattolica del Sacro Cuore, 00168 Rome, Italy; raffaella.zaccaria@policlinicogemelli.it (R.Z.); michele.impagnatiello@policlinicogemelli.it (M.I.); lorenzo.zileridalverme@policlinicogemelli.it (L.Z.D.V.); antonio.gasbarrini@unicatt.it (A.G.)

**Keywords:** sepsis, internal medicine, nutrition, gut microbiota, prevention

## Abstract

Background/Objectives: Sepsis is a life-threatening condition characterized by an imbalanced immune response to infection, posing a significant challenge in hospital settings due to its high morbidity and mortality rates. While much attention has been given to patients in the ICU, uncertainties remain regarding the nutritional management of septic patients in non-intensive wards. This narrative review aims to address these gaps by exploring key aspects of nutritional care in sepsis patients admitted to non-intensive wards. Methods: We examine the pathophysiological mechanisms driving metabolic alterations in sepsis, methods for effective nutritional assessment, and supplementation strategies, including the potential role of specific nutrients. Additionally, we discuss the preventive role of nutrition, with a focus on gut microbiota modulation. Conclusions: By synthesizing the available literature, this review provides evidence-based insights to guide nutritional strategies for managing sepsis in patients hospitalized in non-intensive wards and highlights critical areas for future research.

## 1. Introduction

According to the Sepsis-3 Consensus, sepsis is defined as life-threatening organ dysfunction caused by a dysregulated host response to infection [1]. Septic shock is a subset of sepsis, characterized by profound circulatory, cellular, and metabolic abnormalities. It is clinically identified by the need for vasopressors to maintain a mean arterial pressure of ≥65 mm Hg and a serum lactate level > 2 mmol/L in the absence of hypovolemia.

Several molecular mechanisms are involved in the pathogenesis of sepsis. The most important are immune-inflammatory dysfunction, complement deactivation, mitochondrial damage, and endoplasmic reticulum stress. The severity and prognosis of sepsis do not depend on the pathogenic mechanism, which is common, but on the interaction between host characteristics (such as age, ethnicity, genetic variability, comorbidities, and medications) and pathogen-related factors (including type, load, antimicrobial resistance, and site of infection) [2,3].

Sepsis is a common cause of hospitalization and long-term hospital stays worldwide, with the incidence increasing particularly in Western countries [4]. It frequently affects patients who are elderly and frail, have multiple comorbidities, or suffer from chronic end-stage disease [5]. This subset of septic patients is often admitted to non-intensive wards under three possible scenarios: (1) when vital function support is not required due to the absence of septic shock and/or multiorgan dysfunction; (2) in cases of septic shock and/or multiorgan dysfunction, but ICU admission is deemed futile due to the patient’s age and comorbidities; and (3) immediately after initial stabilization of vital functions in the ICU. Literature data confirm that, even in severe clinical courses, only a minority (20–30%) of septic patients received intensive care. In view of the high mortality rate (between 15 and 56%), sepsis must be suspected, recognized, and treated immediately [6]. Early treatment is associated with a significant improvement in outcomes. Antimicrobial therapy should be initiated as soon as possible in patients with sepsis or septic shock, ideally within the first hour of recognition. Each hour of delay is associated with an additional 4% risk of death [7]. Equally important is the resuscitation of patients with hypovolemia, hypoperfusion, or septic shock using crystalloids and/or vasopressors [8]. Until shock is controlled, enteral or parenteral nutrition is strictly contraindicated. It is only after this initial hyperacute phase that the complex issue of nutritional support arises, which we have addressed in this article. In particular, we focused on the proper management of nutritional therapy in patients with sepsis admitted to non-intensive wards.

## 2. Methods

We conducted a comprehensive screening of articles written in English from January 2000 to August 2024. We searched PubMed, MEDLINE, and Cochraine databases using the following Medical Subject Headings: “sepsis” OR “septic” AND “nutrition”, “metabolism”, “gut microbiota”, “prevention”. We also manually searched the reference lists of the selected articles. We excluded book chapters, conference annals, case reports, animal and pediatric studies, and articles for which no full text was available. After screening, we discussed and agreed on the articles to be included in this review.

We used the GRADE approach to assess the quality of evidence as high, moderate, low, and very low [9].

## 3. Metabolic Alterations in Sepsis

The pathogens causing sepsis in adults can be bacteria, fungi, or viruses. Bacterial etiology (Gram-negative, Gram-positive, or mixed) is still by far the most common [10]. Fungal sepsis follows, mainly due to invasive candidiasis [11]. Fungal sepsis particularly affects immunocompromised patients, post-operative patients, and those undergoing total PN [12]. Only during the initial phase of the COVID-19 pandemic was there an exponential increase (up to 15% of cases) in viral sepsis (SARS-CoV2) [13]. Outside of this exceptional period, viral sepsis is rare. In developed countries, the virus most commonly associated with sepsis is the influenza virus (1–4% of cases) [10,14].

The dysregulated host response to pathogens in sepsis results in a combination of neuronal, endocrine, and immune-inflammatory abnormalities leading to profound micro- and macro-hemodynamic and metabolic consequences [15,16,17]. These abnormalities result in cellular dysfunction and varying degrees of organ dysfunction. The profound and persistent disruption of various metabolic processes observed in sepsis evolves through several phases, which correspond to the ongoing interaction between the pathogen and host (Figure 1).

After a very early phase of abnormal response to the infection, which lasts hours and is characterized by a “metabolic shock”, the body rapidly enters the so-called acute catabolic phase [16]. A combination of cytokine storm, neurohormonal stress response (involving catecholamines, corticosteroids, and glucagon), and insulin resistance play an important role in the acute phase [16,18]. Fasting is common as voluntary food intake ceases, contributing to the metabolic response we will discuss shortly. The body must quickly meet its energy needs, which increase significantly in response to the pathogen’s attack drawing on its reserves. This response is evolutionarily conserved and is essential for short-term survival. However, if it continues for too long, it becomes harmful and causes serious tissue damage. What at first glance looks like an ineffective and harmful response must be interpreted in the light of relatively recent significant improvements in hospital care. In the past, the outcome of sepsis-related organ dysfunction was decided within a few days in an “all or nothing” scenario. Patients either succumbed to the infection or survived, voluntarily resumed eating, and quickly made a full recovery. Today, however, modern hospital care, both inside and outside the ICU, can also enable septic patients with frailty and multimorbidity to survive [19]. During the catabolic phase, one of the most critical responses is the provision of energy substrates to the tissues, despite the sepsis-induced mitochondrial dysfunction and overactivation of the immune-inflammatory response [17]. Glucose becomes the main substrate because glycolysis has the advantage of not requiring oxygen, although it is associated with significantly lower energy production than the Krebs cycle/oxidative phosphorylation even when fasting because insulin levels are elevated. This allows peripheral tissues to utilize glucose as a primary energy source. In addition, hepatic ketogenesis is minimally active during sepsis even when fasting because insulin levels are elevated. This allows peripheral tissues to utilize glucose as a primary energy source [16,18].

Hepatic glycogenolysis supplies glucose for a few hours in times of high demand, as occurs in critical illnesses such as sepsis. As a result, intensive endogenous production of glucose (gluconeogenesis) begins in the liver. The lactate produced during glycolysis is quantitatively the most important precursor of gluconeogenesis, either directly (via the Cori cycle) or indirectly as alanine (via the Cahill cycle, with the transfer of amino groups in the muscle) [20,21]. These cycles lead to net energy loss, as they primarily serve to eliminate waste products. The most important substrates for “de novo” glucose production in the liver are glycerol from increased lipolysis in adipose tissue and amino acids from increased protein catabolism, particularly in muscle [20,21]. The increased availability of amino acids is not only essential for the synthesis of glucose but also for the production of proteins that are crucial for the immune-inflammatory response against pathogens, and thus for recovery from sepsis. This endogenous glucose production is essential for survival and can only be partially suppressed by exogenous glucose or insulin [16].

As already mentioned, although this adaptive response is beneficial in the short term, it has serious long-term consequences. During the increased lipolysis associated with sepsis, the release of fatty acids may exceed energy demands. Excess fatty acids re-esterified into triglycerides can accumulate in the liver and muscle tissue (steatosis) and cause damage, particularly in patients with predisposing conditions such as diabetes and obesity [22]. Protein catabolism is a significant problem in sepsis as it is massive and can lead to a progressive loss of large amounts of LBM, which is associated with a poorer prognosis [23]. Nutritional support during the catabolic phase can reduce but not completely prevent this loss, as it is part of the adaptive response mentioned above [19].

The anabolic phase begins once sepsis has resolved and the stress response subsides. Only then can nutrition effectively counteract the negative protein (and energy) balance, so that the damaged tissue can rebuild.

## 4. Nutritional Assessment and Treatment

The profound metabolic changes that occur in the various phases of sepsis require quantitative and qualitative adaptation of nutritional support during the course of the disease. In addition, patients are often so compromised, especially at the beginning of the disease, that they are unable to eat for several days even if their gastrointestinal tract is functioning. Therefore, a personalized nutritional plan is essential for the proper management of these patients.

Unfortunately, there are few studies on the nutritional management of septic patients, many are of poor quality, and almost all have been conducted in the ICU setting. Since the pathogenic mechanisms underlying sepsis do not differ between patients in and out of the ICU, it is plausible to consider the results of these studies as valid and to use them for the nutritional management of patients admitted to non-intensive wards.

A tailored nutritional program for these patients must always be preceded by a thorough nutritional assessment. Close monitoring is also required to adjust the program according to metabolic changes and resumption of oral feeding. A multidisciplinary approach is crucial, involving collaboration between attending physicians, dietitians and nutritionists, and, if necessary, endoscopists, surgeons, and central venous access team experts.

On admission, it is important to check for pre-existing malnutrition and to assess the risk of malnutrition [24]. For this purpose, the combination of general clinical assessment, laboratory parameters, and validated scores is useful to predict the clinical outcome. Among the laboratory parameters, nutritional biomarkers such as albumin and TTR deserve special mention. Due to its long half-life (about 14–20 days), albumin is not considered useful in assessing acute changes in nutritional status but remains a good marker of chronic malnutrition. The advantage of TTR over albumin is its shorter half-life (2–3 days), so TTR can change more rapidly with changes in nutrient intake [25].

However, neither molecule is a reliable marker of malnutrition in patients affected by inflammatory diseases. The cytokine storm triggered by sepsis stimulates the liver to synthesize inflammatory proteins, such as C-reactive protein, at the expense of albumin and TTR. The latter remains a sensitive prognostic marker for a poor outcome due to malnutrition and severity of illness [25].

Outside the ICU, TTR values > 0.16 g/L are associated with a good prognosis. It has also been shown to be a reliable nutritional marker for refeeding follow-up. Outside the ICU, an increase of >0.04 g/L per week indicates that the ongoing nutritional rehabilitation program is effective [25].

Among the scores, CONUT, PNI, and NRS-2002 are the most commonly used. The NRS-2002 score seems to be the most accurate [26]. The NUTRIC score is also useful, although it has only been validated in ICU patients [27].

In critically ill septic patients admitted to the ICU, indirect calorimetry is strongly recommended to assess REE, which includes basal energy expenditure and diet-induced thermogenesis [28].

Activity-induced energy expenditure should be assessed in recovering patients who resume physical activity. Metabolic changes due to sepsis and common therapies administered to septic patients, such as steroids and vasopressors, are not considered in PE and can only be measured by indirect calorimetry. The Harris–Benedict and Schofield equations are the most commonly used PE for estimating energy expenditure. In non-intensive care settings, where indirect calorimetry is generally not available, physicians are forced to use these imperfect equations [28].

Figure 2 summarizes the nutritional approach tailored to the different phases of sepsis.

In the hyperacute phase of sepsis, nutrition is not necessary, and all efforts are directed towards stabilizing the patient. Intravenous glucose is indicated for the treatment of spontaneous hypoglycemia [29]. After initial stabilization, nutrition can be started in the acute phase of sepsis. The ESPEN guidelines recommend administering hypocaloric nutrition (≤70% of the REE calculated by PE, 15–20 kcal/kg/day) in the first few days of this phase [19,29,30,31].

There is solid physio-pathological evidence for this suggestion. Basal metabolism does not increase by more than 30% compared to normal and decreases more the more severe the sepsis [32]. As described in the previous section, endogenous energy production occurs through massive mobilization of caloric reserves (adipose tissue, muscle, glycogen). The authors of the ESPEN guidelines formulated their recommendation based on the results of RCTs involving non-malnourished critically ill patients admitted to the ICU due to conditions of various causes (medical, surgical, or traumatic). In the large multi-center study by Arabi et al., approximately 30% to 35% of the 894 patients were admitted for severe sepsis [33]. Patients were randomized to receive permissive underfeeding (40% to 60% of calculated caloric requirements) or standard enteral feeding (70% to 100%) for up to 14 days while maintaining similar protein intake. Age (50.2 ± 19.5 and 50.9 ± 19.4 years) was similar between groups. Ninety-day mortality was not significantly different (27.2% and 28.9%, respectively). No serious adverse events were reported. No significant differences were found in feeding intolerance, diarrhea, ICU-acquired infections, or ICU or hospital LOS. The meta-analysis by Tian et al. evaluated eight RCTs, including the above-mentioned study by Arabi et al., with a total of 1895 patients [34]. No significant difference in mortality was found between the low- and high-energy groups (RR 0.90; 95% CI 0.71–1.15; *p* = 0.40). The risk of infection and gastrointestinal intolerance also did not differ. The low-energy subgroup, which received 33.3 to 66.6% of target energy, was associated with significantly lower mortality than the high-energy group (RR 0.68; 95% CI 0.51–0.92; *p* = 0.01). Gastrointestinal intolerance also improved significantly. The moderate-quality results of this meta-analysis need to be confirmed by targeted RCTs on acute nutrition in septic patients, which are currently unavailable. The problem is even greater in malnourished patients as they were excluded from the above studies. In the absence of specific evidence, the ESPEN guidelines recommend a higher calorie target (25 kcal/kg/day) for malnourished patients (BMI < 17 m^2^), as they do not have sufficient metabolic reserves to meet energy requirements [19].

This goal should be achieved gradually, as the risk of refeeding syndrome and associated life-threatening complications is quite high in malnourished patients undergoing prolonged fasting [19,29]. The recommended amounts of lipids and glucose are 0.7–1.5 g/kg/day and 1–1.5 g/kg/day, respectively [18,19,30]. Protein loss is considerable from the first day of illness, and supplementation is often inadequate compared to actual requirements. The recommended protein intake is about 1.0 g/kg/day (ranging between 0.8 and 1.3 g/kg/day) and should be administered as soon as possible after admission [18,19,31]. This recommendation is also derived from moderate-quality RCTs in critically ill patients admitted to the ICU for diseases of various etiologies, with sepsis representing a minority [34]. The calculation of nitrogen balance may be a useful tool in non-intensive wards to assess whether protein intake is sufficient to meet the body’s needs in the different metabolic phases of sepsis. The aim is to reverse a negative nitrogen balance, as its improvement is associated with a better outcome [35].

A standard carbohydrate-containing nutrition may exacerbate the metabolic and immunologic dysfunction associated with sepsis. A ketogenic diet, which is high in fat and very low in carbohydrates, mimics the metabolic effects of fasting. This process involves the breakdown of triglycerides to form ketone bodies, which provide ATP as an alternative energy source during calorie deprivation. Two recent small clinical trials are available. In a pilot study by McNelly and coll., 29 mechanically ventilated adults with multi-organ failure (median age 52.0, 45.5–58.5 years) were randomized [36]. The ketogenic diet was feasible, safe, and well-tolerated. It was associated with fewer hypoglycemic events and fewer insulin requirements. Rahmel et al. conducted an open-label trial involving 40 ICU patients with sepsis (66.9 ± 13.8 years) [37]. No major adverse events or metabolic side effects were observed. None of the patients in the ketogenic diet group required insulin treatment, while insulin dependency in the control group ranged from 35% to 60% (*p* = 0.009). No difference was found in about 30-day survival, but ventilation-free, vasopressor-free, dialysis-free, and ICU-free days were significantly higher in the ketogenic group. Larger RCTs are needed to confirm these very low-quality evidence results, which suggest that a ketogenic diet may have potential benefits for humans.

After the acute phase of sepsis, it is important for the patient to receive a higher intake of calories (25–30 kcal/kg/day) and protein (1.2–2.0 g/kg/day). This increase is crucial to minimize further loss of LBM and to promote early mobilization [18,19,30].

After the resolution of sepsis, the patient enters the recovery phase. At this point, one of the main goals is to regain autonomy. While this can sometimes occur spontaneously, early and progressive physical therapy is often necessary, especially for elderly and frail patients. To support physical activity and metabolic changes during recovery, calorie and protein intake should be increased (40–45 kcal/kg/day—protein 2.0 g/kg/day) [19].

Upon discharge, the patient should receive a detailed nutritional plan. Given the high prevalence of inadequate dietary intake due to severe sepsis-related impairment, pre-existing frailty, and multimorbidity, oral nutritional supplements should be prescribed for 3–12 months [19,30]. The quality of evidence for this recommendation is low. In fact, it is based on clinical studies conducted in different settings and populations, not specifically designed for post-sepsis patients [19,38]. A systematic review by Cawood et al. including 36 RCTs (3790 patients, mean age 74 years; 83% of studies on patients > 65 years) suggests that high-protein ONS (energy > 20% from protein) can improve long-term outcomes in older adults. These benefits include reduced complications, such as pressure ulcers and fractures (OR 0.68; 95% CI 0.55–0.83), fewer hospital readmissions (OR 0.59; 95% CI 0.41–0.84), improved grip strength (1.76 kg; 95% CI 0.36–3.17), and weight gain (*p* < 0.001) [38].

Post-sepsis patients should be closely monitored through regular outpatient visits to personalize nutritional interventions and facilitate a progressive recovery of weight and LBM.

### 4.1. Feeding Route

As previously mentioned, patients hospitalized for sepsis or septic shock should only receive nutritional support after successful resuscitation [19,29].

For conscious, non-intubated patients without gastrointestinal contraindications, physiological nutrition can be resumed orally [30]. Whenever possible, the enteral route should be preferred as it preserves intestinal integrity and permeability, while also contributing to the modulation of the inflammatory response and insulin resistance. It is important to carefully monitor the amount of prescribed food consumed by patients re-fed via an oral diet. ONS should be prescribed for patients who are not eating enough. If, despite ONS, the patient still fails to meet the pre-established caloric and protein targets, EN should be initiated [30].

In patients unable to receive an oral diet (with or without ONS), the feeding route should be enteral and/or parenteral. EN is not only the most physiological feeding route but also more cost-effective than PN. EN should be preferred unless there are clear contraindications to the enteral feeding route [18,19,29,30]. According to the 2023 ESPEN guidelines on clinical nutrition in intensive care, EN should be started early, provided there are no absolute contraindications. Initially, 20–50% of the complete nutritional requirement should be administered to assess tolerance and achieve a trophic effect on the gastrointestinal system. If well tolerated, EN should be progressively increased until the optimal nutritional goal is reached [30]. Based on the available literature, this approach is specifically applicable to patients with sepsis, particularly those with septic shock. A concern is that impaired splanchnic perfusion may impose an additional workload on the intestine of these patients, increasing the risk of non-obstructive intestinal ischemia or necrosis. The large multi-center RCT NUTRIREA-2, involving critically ill adults with shock (60% septic shock, mean age 66 years), did not find a significant difference in 28-day mortality between early isocaloric EN and PN (37% of 1202 patients in the enteral group versus 35% of 1208 patients in the parenteral group) [39]. The incidence of ICU-acquired infections was also similar between groups (14% versus 16%). However, a significantly greater risk of digestive complications (vomiting, diarrhea, bowel ischemia, and acute colonic pseudo-obstruction) was observed in the EN group. The NUTRIREA-3 study enrolled 3044 patients with similar characteristics to those in the NUTRIREA-2 study (mean age 66 years, admitted to ICU for shock, 60% septic shock) [40]. They were randomly assigned to early nutrition with either low or standard calorie and protein targets (6 kcal/kg per day and 0.2–0.4 g/kg per day protein versus 25 kcal/kg per day and 1.0–1.3 g/kg per day protein, respectively). Compared with standard calorie and protein targets, early calorie and protein restriction did not decrease mortality but was associated with faster ICU discharge and fewer complications (vomiting, diarrhea, bowel ischemia, and liver dysfunction). In a recent meta-analysis of five RCTs and 10 non-randomized studies, including a total of 4166 patients, low-quality evidence suggested that early EN may be a safe and effective intervention in critically ill patients with sepsis or septic shock [41]. The authors found no significant difference in mortality between patients receiving early EN and no or delayed EN, in either RCTs or non-randomized studies. The early EN group could require fewer days of MV and had lower SOFA scores during follow-up, although a higher frequency of diarrhea was observed. The risk of gastrointestinal complications with early EN seems to be related to the severity of septic shock. In fact, EN tolerance is inversely related to the maximum dose of norepinephrine administered [42]. In conclusion, the “less is more” strategy proposed by the NUTRIREA-3 study is certainly preferable in patients with severe septic shock. For those without shock or on low vasopressor doses, early EN with the rapid achievement of the protein–calorie targets could be a valid approach [19].

PN should be prescribed if, after 3 days, EN is not tolerated or sufficient. It should provide 50% of the predicted or measured energy requirement, pending a potential resumption of EN [29].

### 4.2. Micronutrients and Electrolytes

Micronutrients are vitamins and trace elements required by the body in small amounts. They play a crucial role in various enzymatic reactions, such as ATP production, antioxidant activity, and immune defense. In the absence of adequate food intake, particularly in malnourished patients, individuals with sepsis or septic shock may rapidly deplete several micronutrients. The most common deficiencies include thiamine, vitamin C, vitamin D, and selenium [43]. Restrictive feeding during the acute phase may exacerbate micronutrient deficiencies. The most common electrolyte abnormalities observed in critically ill patients involve magnesium, potassium, calcium, phosphorus, and sodium [44].

The rationale for considering supplementation of these elements in critically ill patients, particularly those with sepsis, is twofold. First, it helps prevent refeeding syndrome. Second, it can favorably modulate pathophysiological processes involved in sepsis, where, as mentioned, immune-inflammatory and dysmetabolic aspects play a crucial role. Many of these micronutrients possess antioxidant, anti-inflammatory, and immunomodulatory properties. Other specific functions are briefly discussed. Vitamin C is a cofactor in the production of endogenous amines. A small study of patients with septic shock demonstrated that vitamin C infusion significantly increased indicators of norepinephrine synthesis and reduced the required dosage of exogenous norepinephrine compared to placebo [45]. Thiamine (vitamin B1) is a cofactor in many mitochondrial enzymatic reactions. It is considered a mitochondrial resuscitator that could mitigate organ injury, particularly renal injury, in septic shock [46].

In critically ill patients who resume feeding, subclinical micronutrient and electrolyte deficiencies may manifest as refeeding syndrome [29]. During refeeding after prolonged starvation, increased intracellular uptake of several electrolytes (phosphate, potassium, and magnesium) and micronutrients (thiamine, other B vitamins, and trace elements) occurs. Due to reduced stores, concentrations rapidly decline in the body, increasing the risk of life-threatening complications, including cardiopulmonary, hematologic, and neurological dysfunction, arrhythmias, severe muscle weakness, lactic acidosis, and water retention. Diagnosing refeeding syndrome can be challenging. A most commonly used criterion is a decrease in phosphate levels by at least 0.16 mmol/L to below 0.65 mmol/L during refeeding [29]. In patients who develop refeeding hypophosphatemia, temporary caloric restriction, rather than electrolyte correction, is effective in reducing mortality [47]. While concrete evidence is lacking, a gradual increase in caloric intake after resuming nutrition during the acute phase, combined with sufficient micronutrient and electrolyte supplementation, seems prudent to prevent refeeding syndrome. Optimal dosages remain unclear, as plasma concentrations do not accurately reflect micronutrient stores or redistribution during inflammation. Micronutrient infusion can be safely discontinued once the patient receives sufficient macronutrients (via food or EN). It is important to note that, unlike standard commercial EN formulations, PN does not contain micronutrients. Presumed optimal intakes for critically ill patients, such as those with sepsis, can be found in the ESPEN micronutrient guidelines [48]. 

Vitamin C is the most extensively studied micronutrient. A recent meta-analysis by Wen and coll. on 24 RCTs demonstrated that IV administration of vitamin C to patients with sepsis was associated with a trend toward improved mortality (RR 0.86; 95% CI, 0.74–1.01; *p* = 0.06) [49]. SOFA scores of patients with sepsis significantly improved after vitamin C treatment (RR; 0.26; 95% CI; 0.09–0.42; *p* = 0.002). In 14 of these 24 studies, vitamin C was given with hydrocortisone and thiamine, following the promising results of the famous “Marik protocol” for the resuscitation of ICU septic patients [50]. In two other studies, vitamin C and thiamine were co-administered. The network meta-analysis by Safabakhsh and coll. excluded studies in which vitamin C was co-administered with hydrocortisone [51]. Vitamin C alone (eight RCTs), but not vitamin C co-administered with thiamine (2 RCTs), was associated with a significant improvement in short-term mortality (RR 0.81; 95% CI 0.67–0.99).

As reported by Li et al., selenium infusion (eight studies including five RCTs) did not affect mortality [52]. Interestingly, patients treated with selenium had a shorter duration of vasopressor therapy, shorter ICU and hospital LOS, and a lower incidence of ventilator-associated pneumonia. Like selenium, IV thiamine did not significantly affect the prognosis of septic patients [51]. However, two recent studies suggest interesting perspectives on the use of thiamine in sepsis. In a post hoc analysis of two randomized trials involving 158 patients with septic shock (median age 70, 60–79 years), thiamine administration was associated with significantly lower in-hospital mortality only in the thiamine-deficient group (thiamine < 8 nmol/L, 16% in the treatment group versus 59% in the placebo group, adjusted OR 6.84; 95% CI 1.54–30.36) [53]. Zhang et coll. retrospectively enrolled 11,553 ICU patients with sepsis, of whom 1536 (59, 49–68 years) received and 10,017 (68, 56–79 years) did not receive thiamine supplementation. After controlling for potential confounders, the thiamine-supplemented group had a significantly lower mortality risk than the non-supplemented group. The hazard ratio of ICU mortality for the supplemented group was 0.80 (95% CI 0.70–0.93) [54].

The effect of vitamin D treatment in septic patients was evaluated in four RCTs. No significant effect on mortality, ICU and hospital LOS, MV, or vasopressor duration was reported [51]. A recent study by Ashoor et coll. enrolled 80 patients with sepsis requiring MV and evidence of vitamin D deficiency. They were randomly assigned to receive enteral vitamin D supplementation of 50,000 or 5000 IU. The high dose (50,000 IU) showed a significant improvement in procalcitonin and the SOFA score and a decrease in ventilator-associated pneumonia and hospital LOS [55].

### 4.3. Other Disease-Specific Nutrients

There are literature data on other molecules that have the potential to favorably modulate the pathophysiology of sepsis. The most studied are omega-3 fatty acids, glutamine, and arginine.

Omega-3 polyunsaturated fatty acids (PUFAs), including eicosapentaenoic acid and docosahexaenoic acid, provide energy and nutrients to the body, regulate lipid metabolism, play an anti-thrombotic role, and modulate immunity and inflammation [18]. In a meta-analysis of 25 RCTs with 1903 participants, omega-3 PUFA supplementation in adult patients admitted to the ICU for sepsis was associated with lower mortality compared to the control group. Lower mortality was noted particularly in the PN subgroup, while the addition of omega-3 PUFAs did not significantly affect mortality in the EN subgroup [56]. A recent network meta-analysis of 28 RCTs substantially confirmed these results [57].

Glutamine is the most abundant amino acid in the human body. It is an essential nutrient for immune cells (lymphocytes, macrophages, and neutrophils) and enterocytes. It also stimulates nucleotide synthesis and possesses antioxidant properties. In sepsis, glutamine deficiency may arise from a combination of reduced food intake, increased immune activity, and hypercatabolism [58]. A meta-analysis of 47 RCTs involving 6198 critically ill adult patients admitted to the ICU for various diseases showed that glutamine supplementation did not affect mortality [59]. The route of administration (enteral or parenteral) and dosage (high, moderate, or low) had no impact on mortality in medical, surgical, or trauma patients. Currently, there are no specific clinical trials in septic patients.

l-arginine is a conditionally essential amino acid that is substantially decreased in patients with sepsis [60]. Due to its immunomodulatory, metabolic (protein synthesis), and vascular properties (stimulation of nitric oxide synthesis), a role for arginine supplementation in sepsis has been postulated [60]. However, prolonged intravenous l-arginine administration does not improve local perfusion, organ function, or protein metabolism in patients with septic shock [61].

Among the nutrients discussed in this subsection, only the literature on ω-3 PUFAs showed moderate-quality evidence.

Table 1 summarizes the results of studies available to date on the effects of micronutrients and other specific nutrients in patients with sepsis.

### 4.4. Controversies Surrounding Non-Caloric Non-Protein Nutritional Supplements

The numerous studies published to date on the effect of micronutrients and disease-specific nutrients in sepsis have several shortcomings. Firstly, it is difficult to compare the results of different studies due to the extreme variability in dose and duration of treatment and routes of administration. Furthermore, especially as seen with vitamin C and thiamine, these have been variously combined with each other and with corticosteroids. This makes it difficult to understand the actual contribution of the various treatment components to the outcome. Finally, many of these studies are characterized by small sample sizes. Therefore, to date, there is insufficient evidence to recommend the use of these dietary supplements in septic patients. Regarding micronutrients, their supplementation seems useful in the first days after fasting and in malnourished patients, and also to avoid refeeding syndrome.

## 5. Sepsis Prevention: Role of Nutritional Status, Diet, and Gut Microbiota

Quantitative and qualitative nutritional abnormalities can significantly contribute to sepsis risk. In less developed regions, malnutrition is common in both children and adults. In developed countries, it is typical of frail older adults. Malnutrition is associated with an increased risk of infection and sepsis, as well as more severe outcomes [24]. Chronically insufficient dietary intake leads to immunodeficiency. In fact, excessive protein catabolism, abnormal glucose and lipid metabolism, and deficiencies of several vitamins and micronutrients impair the function of both the innate and adaptive immune systems. In addition, malnutrition leads to mucosal damage, which facilitates pathogen invasion [15,44]. Conversely, an infection can lead to decreased nutritional status and immunosuppression, creating a vicious cycle that favors secondary infections [64,65]. Excess energy intake resulting in obesity has been linked to an elevated risk of infection and sepsis [65,66]. In a large cohort, individuals with BMI ≥ 40 had a significantly higher risk of sepsis compared to those with normal BMI. Several pathophysiological mechanisms have been hypothesized. Increased fatty tissue promotes a persistent inflammatory state characterized by elevated production of cytokines, such as IL-6 and TNF-alpha [65]. Furthermore, adipocytes express Toll-like receptors, which are responsive to endotoxin. Oxidative stress and elevated lipid concentrations typical of obesity can lead to apoptosis and endothelial dysfunction [65]. The specific composition of nutrients in the diet may also play a pivotal role in sepsis. WD, one of the most common dietary patterns in Westernized nations, is characterized by a high caloric content and is enriched in animal proteins, saturated fats, simple sugars, and ultra-processed foods while being deficient in fiber, fruits, and vegetables. In a US national cohort of 21,404 adults, a dietary pattern characterized by fried foods, processed meats, and sugar-sweetened beverages was independently associated with a long-term risk of sepsis [67]. In animal models of LPS-induced sepsis, WD exacerbates disease severity and outcomes compared to a standard high-fiber diet. Mice fed WD show higher baseline inflammation and signs of sepsis-associated immunoparalysis (impaired immune cell migration and neutrophil function) compared to mice fed standard high-fiber chow [68]. Population-based studies have found that WD, unlike diets rich in fruits and vegetables, is associated with key mediators of sepsis, such as inflammation and endothelial cell activation [69].

This association is similar to that observed in obesity, of which WD is the most common dietary pattern [69]. The gut microbiota may play a crucial role in the interplay between diet patterns and sepsis. The gut microbiota is a complex ecosystem within the body that exerts a significant influence on innate and adaptive immunity, metabolism, intestinal growth, and intestinal permeability. Dietary regimes can substantially modify the composition and functionality of the gut microbiota [70,71]. For instance, the Mediterranean diet has a positive impact on the gut microbiota and human health. A higher amount of *Faecalibacterium* spp. is associated with greater production of anti-inflammatory molecules and short-chain fatty acids, the latter essential for intestinal trophism [70]. A significant decrease in gut microbiota diversity has been reported in WD compared to other diets. In WD, a different microbial composition, with a prevalence of *Bacteroides*, and lower fiber intake cause a decreased production of short-chain fatty acids [70]. The numerous additives, preservatives, and emulsifiers also influence the gut microbiota in WD. For example, carrageenan can induce gut inflammation and disrupt the mucus layer, leading to a negative shift in the gut microbiota. Artificial food colorings capable of modifying sulfur homeostasis, and some preservatives, such as sodium nitrate present in processed meat, can also alter gut microbiota composition. WD, by profoundly modifying the gut microbiota, causes persistent immuno-inflammatory dysfunction and altered intestinal trophism, which can predispose patients to sepsis. Since diet can modify the gut microbiota, nutritional interventions that consider not only protein–calorie requirements but also food quality could be beneficial in restoring an altered gut microbiota [72]. Probiotics, defined as “live microorganisms that, when administered in adequate amounts, confer a health benefit on the host”, and prebiotics, which are substrates selectively utilized by host microorganisms conferring a health benefit, may support dietary modification in restoring a “physiological” gut microbiome.

However, to date, there are no significant studies evaluating the impact of correcting quantitative and qualitative nutritional abnormalities on the risk of sepsis and its outcome, despite the interesting premises discussed above.

## 6. Conclusions

Sepsis is a common cause of admission to non-intensive wards. These patients may not require advanced life support or may have septic shock/multiple organ dysfunction syndrome, but their underlying medical condition may be so poor that ICU admission is considered futile. Septic patients are often transferred from the ICU to general wards after clinical stabilization during the acute course of the disease.

Sepsis is characterized by significant metabolic derangement due to the interaction between the pathogen and the human host. Early in the disease, starvation is common. A prompt assessment of nutritional status and needs is essential upon presentation and throughout the course of sepsis, as nutritional requirements can vary significantly between the acute and recovery phases. Both underfeeding and overfeeding are associated with poor outcomes. Due to the limitations of PE in critically ill patients, the use of indirect calorimetry is recommended for an accurate measurement of nutritional requirements. However, this method is generally not feasible in non-intensive wards. Many recommendations in sepsis nutrition guidelines are based on studies involving critically ill patients admitted to the ICU for conditions other than sepsis, including surgery and trauma. In addition, the few sepsis-specific studies, which are often small and/or retrospective, have primarily focused on patients with septic shock. While the available literature demonstrates substantial progress in this field, it also underscores the need for further research to define the optimal nutritional approach for patients with sepsis/septic shock admitted to non-intensive wards.

Large prospective interventional RCTs are warranted to determine key factors such as timing, energy requirements, quantity and distribution of calories, route of feeding, and potential benefits of supplementation with micronutrients, electrolytes, and other pharmaco-nutrients.

Finally, available data suggest that hospitalization for sepsis presents an opportunity to conduct a comprehensive review of the patient’s dietary habits and nutritional needs upon discharge. Future studies investigating the effects of dietary modifications or probiotic/prebiotic supplementation on sepsis risk and outcomes are welcome.

## Figures and Tables

**Figure 1 nutrients-16-03985-f001:**
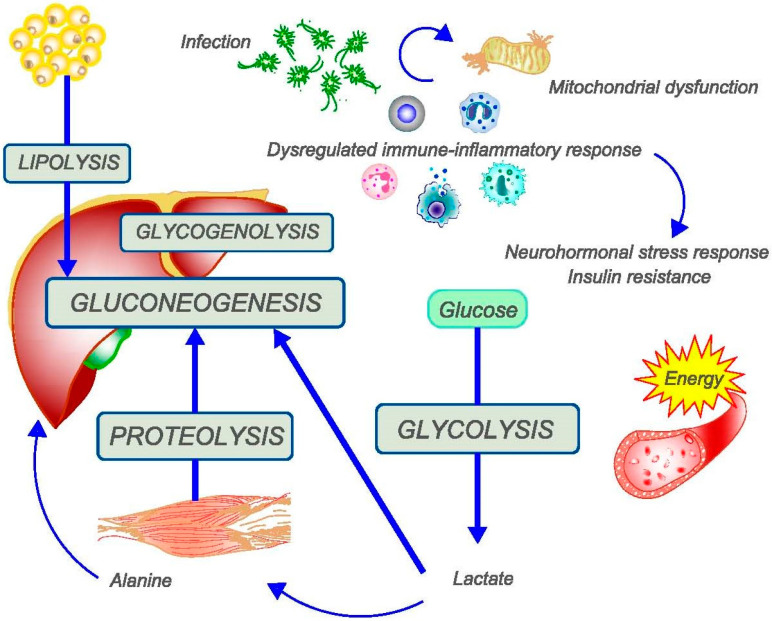
Schematic representation of the complex metabolic alterations observed in the acute phase of sepsis. Glucose becomes the primary substrate, as glycolysis has the advantage of not requiring oxygen, although energy production is significantly lower than in the Krebs cycle. Hepatic ketogenesis is suppressed by increased insulin levels. This allows peripheral tissues to utilize glucose as a primary energy source. Hepatic glycogenolysis provides glucose for a short time. This is followed by intensive endogenous glucose production (gluconeogenesis) in the liver from lactate, amino acids derived from increased protein catabolism, and glycerol derived from increased lipolysis.

**Figure 2 nutrients-16-03985-f002:**
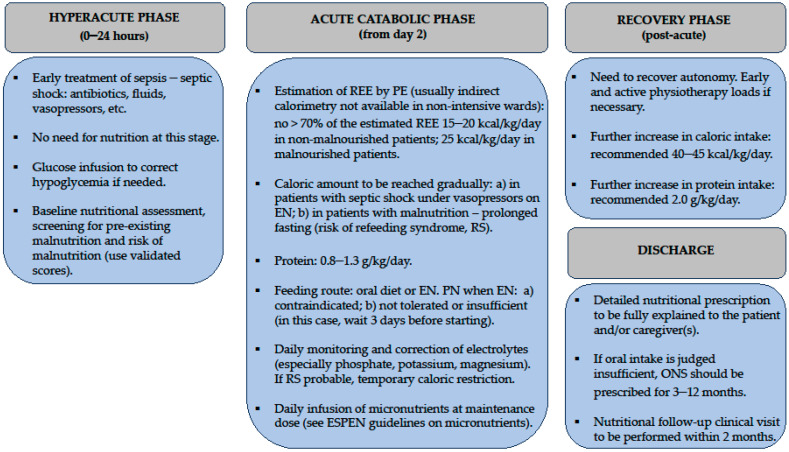
Nutritional management of patients admitted to non-intensive wards for sepsis.

**Table 1 nutrients-16-03985-t001:** Summary of clinical studies and outcomes related to micronutrient and other disease-specific supplementation in patients with sepsis.

Nutritional Intervention	Rationale	Results of Clinical Studies	Dose and Duration	Quality of Evidence
Magnesium Noormandi et al., 2020 [62]	- Reduced level in pts with ICU pts. - Involved in storage and transfer of energy, protein, and nucleic acid synthesis, inflammation, other electrolyte balance, and hemostasis.	- No significant effect on mortality. - Significantly shorter time to lactate clearance and ICU LOS.	IV to maintain serum level around 3 mg/dL for 3 days	Very low (one RCT on severe sepsis)
Selenium Li et al., 2019 [52] Safabakhsh et al., 2024 [51]	- Reduced level in pts with sepsis. - Antioxidant properties involved in the immune response and regulation of thyroid hormones.	- No significant effect on mortality, incidence of renal failure, secondary infection, or duration of MV. - Significantly shorter duration of vasopressor therapy, ICU and hospital LOS, and lower incidence of ventilator-associated pneumonia.	IV, different dose/scheme, mean duration 14 days	Low (meta-analysis of eight studies, five RCTs)
Zinc Kim et al., 2024 [63]	- Increased need. - Involvement in the immune system and cytokine production.	- No significant effect on mortality or ICU LOS.	EN, three doses: <15 mg, 15–50 mg, ≥50 mg, during ICU stay	Very low (one retrospective study)
Vitamin C Wen et al., 2023 [49] Safabakhsh et al. 2024 [51]	- Reduced level in pts with sepsis, - Regulation of immune system and of cytokine homeostasis; anti-inflammatory and antioxidant properties.	- A trend toward improvement of overall and 28-days mortality. Statistically significant when considering eight studies with vitamin C alone [51]. - Significant improvement of SOFA scores. - Significantly shorter duration of MV and vasopressors.	IV, mean dose 6 g/day, mean duration 4 days Used alone [51] (eight studies) or combined with thiamine and/or corticosteroids [49]	Moderate (meta-analysis of 24 RCTs)
Vitamin D Safabakhsh et al., 2024 [51] Ashoor et al., 2024 [55]	- Reduced level in pts with sepsis. - Regulation of immune system and inflammatory response to infections.	- No significant effect on mortality, ICU and hospital LOS, or duration of MV and vasopressors. - A recent study [55] showed a high dose (50,000 IU) was associated with significant improvement in procalcitonin and SOFA score and a decrease in ventilator-associated pneumonia and hospital LOS.	Enteral route, single dose of vitamin D3 (high or low) or cholecalciferol IV, high single dose of calcitriol	Low (four RCTs)
Thiamine Safabakhsh et al., 2024 [51]	- Reduced level in ICU pts with septic shock. - Essential for normal mitochondrial function (aerobic respiration); antioxidant properties.	- No improvement in mortality. - In a post hoc analysis of a thiamine-deficient cohort (thiamine < 8 nmol/L), administration was associated with higher OR of being alive and RRT-free at hospital discharge.	IV, 400 mg, mean duration 5 days	Low (five RCTs, all on septic shock)
Omega-3 PUFA Tseng et al., 2024 [57]	- Anti-inflammatory activity, modulation of immune system, and organ/tissue protection.	- Only high-dose fish oil associated with significant improvement in mortality and inflammatory markers. - Any dose associated with significant improvement in organ failure severity, ICU and hospital LOS, and MV.	IV, PN containing fish oil (high dose when ≥0.5 g/kg/day IV), mean duration 7 days	Moderate (meta-analysis of 28 RCTs)
Glutamine Liang et al., 2024 [59]	- Reduced level in pts with sepsis. - Amino acid involved in normal trophism of enterocytes and normal function of immunologic system.	- No significant effect on mortality, ICU LOS, or infectious complications. - Significant reduction in hospital LOS.	EN containing glutamine 0.3–0.5 g/Kg/day, mean duration 7 days	Low (meta-analysis of RCTs in ICU pts with different acute diseases)
l-arginine Luiking et al., 2020 [61]	- Reduced level in pts with sepsis. - Amino acid involved in cellular regeneration, immune function, protein synthesis, and NO synthesis.	- No improvement in local perfusion or organ function despite an increase in whole-body NO synthesis.	IV, 1.2 μmol·kg^−1^·min^−1^, 3 days	Very low (one RCT on septic shock)

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
