# Peer review of "Nutritional Strategies for the Treatment and Prevention of Sepsis Outside the Intensive Care Unit"

_nutrients, 2024, doi:10.3390/nu16233985_

Round 1
Reviewer 1 Report
Comments and Suggestions for Authors
This article is mainly about nutritional management during sepsis. The article is highly relevant to this field.
This is a different angle over sepsis. The authors have done a comprehensive review over the nutritional
strategies during sepsis. This is an important aspect in the management of sepsis.
1. The references are all appropriate, but the number is a little small, and can be supplemented appropriately.
2. Figures and diagrams are OK.
Overall, The review is well written and can be published in current form.
Author Response
Comments 1. “The references are all appropriate, but the number is a little small, and can be supplemented appropriately”.
Response 1. Thank you very much for your comments. As suggested, we supplemented appropriately the references.
Reviewer 2 Report
Comments and Suggestions for Authors
The narrative review would be of interest, while there seemed to be points that might lead to some improvement of the paper’s contents.
1. Title and overall concept; the consideration of studies in internal medicine is a little described; however, sepsis always occurs regardless of departments. For instance, even though patients who hospitalize in Surgical Department, we see sepsis based on the underlying diseases in internal medicine. The definition of internal medicine department is ambiguous. Does sepsis in patients with dialysis departments belong to those in internal medicine departments, or not? More reasonable statements might be necessary for the overall concept of this paper.
2. Abstract; the more concrete and detailed findings could be described.
3. Methods; the rational and method of narrative review could be detailed. The criteria and validation to select references could be further detailed.
4. Methods and overall text; the consideration of ‘evidence’ level of cited references could be described.
5. Text; clinical characteristics of patients (i.e., mean age, the severity of sepsis) in each study could be added when citing references and describing the results of studies.
6. Text; the other nutritional biomarkers such as albumin and transthyretin might be more described.
7. Text; it might be better to list the nutrients described in the text in a table and to summarize their effects on sepsis.
8. Row 37; adding the number of references to the end of sentence would be better.
9. Row 45; adding the number of references to the end of words ‘high mortality rate’ would be better. The rates might be also described.
10. Row 57; what is the pathogens? The types and spaces of pathogens might be detailed.
11. Relating to the above point, how about fungi?
Comments on the Quality of English Language
Minor editing of English language required, but not so bad.
Author Response
Comments 1. “Title and overall concept; the consideration of studies in internal medicine is a little described; however, sepsis always occurs regardless of departments. For instance, even though patients who hospitalize in Surgical Department, we see sepsis based on the underlying diseases in internal medicine. The definition of internal medicine department is ambiguous. Does sepsis in patients with dialysis departments belong to those in internal medicine departments, or not? More reasonable statements might be necessary for the overall concept of this paper”.
Response 1. Thank you for pointing this out. We agree with this comment. We changed titled and text as suggested.
Comments 2. “Abstract; the more concrete and detailed findings could be described”.
Response 2. As suggested, we changed the abstract.
Comments 3. “Methods; the rationale and method of narrative review could be detailed. The criteria and validation to select references could be further detailed”.
Response 3. As suggested, we provided further details about rationale, methods and criteria and validation to select references.
Comments 4. “Methods and overall text; the consideration of ‘evidence’ level of cited references could be described”.
Response 4. As suggested, we described the consideration of ‘evidence’ level of cited references.
Comments 5. “Text; clinical characteristics of patients (i.e., mean age, the severity of sepsis) in each study could be added when citing references and describing the results of studies”.
Response 5. As suggested, we added clinical characteristics of the patients in the studies cited in the text.
Comments 6. “Text; the other nutritional biomarkers such as albumin and transthyretin might be more described”.
Response 6. As suggested, we described other nutritional biomarkers such as albumin and transthyretin.
Comments 7. “Text; it might be better to list the nutrients described in the text in a table and to summarize their effects on sepsis”.
Response 7. As suggested, we created a new table to describe and summarize the effects of the nutrients described in the text.
Comments 8. “Row 37; adding the number of references to the end of sentence would be better”.
Response 8. Row 37 - As suggested, we added the number of references to the end of the sentence.
Comments 9. “Row 45; adding the number of references to the end of words ‘high mortality rate’ would be better. The rates might be also described”.
Response 9. Row 45 - As suggested, we added the number of references to the end of words ‘high mortality rate’ and we described the rates reported in this study.
Comments 10. “Row 57; what is the pathogens? The types and spaces of pathogens might be detailed.”.
Response 10. Row 57 - As suggested, we specified types and spaces of pathogens.
Comments 11. “Relating to the above point, how about fungi?”.
Response 11. As suggested, we better specified about the gut microflora and its composition, enlarging the explanation to microorganisms other than bacteria such as fungi.
Minor editing of English language required, but not so bad.
As suggested, English language was edited.
Reviewer 3 Report
Comments and Suggestions for Authors
In this narrative review, the authors present literature data on the impact of nutritional management in sepsis for patients admitted to the Internal Medicine department, primarily drawing from strategies used in the ICU. The topic is interesting as it addresses clinical guidance for septic patients, a population with unique challenges and high mortality rates. However, there are several significant issues with the manuscript that need to be addressed. My recommendations are as follows:
1. Abstract:
Revise the abstract to read as follows:
"Sepsis is a life-threatening condition characterized by an imbalanced immune response to infection, posing a significant challenge in hospital settings due to its high morbidity and mortality rates. While much attention has been given to patients in the Intensive Care Unit (ICU), uncertainties remain regarding the nutritional management of septic patients in other departments, particularly in Internal Medicine. This narrative review aims to address these gaps by exploring key aspects of nutritional care in sepsis patients admitted to the Internal Medicine department. We examine the pathophysiological mechanisms driving metabolic alterations in sepsis, methods for effective nutritional assessment, and supplementation strategies, including the potential role of specific nutrients. Additionally, we discuss the preventive role of nutrition, with a focus on gut microbiome modulation. By synthesizing the available literature, this review highlights critical areas for future research and provides evidence-based insights to guide nutritional strategies for managing sepsis in Internal Medicine patients."
Comment: The aim of the review should be placed at the end of the abstract.
2.Lines 32-35:
Please rephrase this paragraph as follows:
"The severity and prognosis of sepsis are influenced by multiple factors, including host characteristics (such as age, race, genetic variability, comorbidities, and medications), as well as infection-related factors, including the type, location, pathogen load, and antimicrobial resistance."
3.Lines 48-49: specify the mortality rates
4.Line 65:
Incorporate a figure legend for figure 1.
5. Line 161:
Present a brief comment on the results of the studies discussed to provide a critical interpretation of the findings.
6. Line 165:
Remove the phrase: “(see section 3.2 for more details on refeeding syndrome).”
7. Line 216:
There seems to be missing information regarding the NUTRIREA-2 trial, which may have been accidentally omitted, as the trial is discussed in the following section.
8. Line 220:
Clarify what is meant by “either testing”. If the authors are referring to tolerance, the following revision may be used (lines 218-221):
"According to the 2023 ESPEN guidelines on clinical nutrition in the ICU, EN should be initiated early, provided there are no absolute contraindications. Initially, 20-50% of full nutritional requirements should be administered to either assess tolerance or achieve a trophic effect on the gastrointestinal system. If well tolerated, EN should be progressively increased until the optimal nutritional target is reached."
9.Nutritional Assessment and Treatment:
1.Initially, the authors should present the potential benefits of nutrition in sepsis within the main text. Following this, at the end of the section, a brief subsection should be included that discusses the controversies surrounding nutritional supplements, drawing on evidence from various studies.
2.Incorporate the results of recent studies and meta-analyses to strengthen this section:
- On the role of Vitamin C in sepsis (PMID: 36743822).
- On the role of Thiamin in sepsis prognosis (PMID: 36259460).
3.General Comment:
It would be helpful to include a table summarizing the clinical studies and outcomes related to micronutrient and other disease-specific supplementation. Besides, this table could present additional data on micronutrients not mentioned in the main text, such as Zinc (PMID: 39275159).
10.Lines 423-427:
The authors should revise this part to avoid repeating that most data are derived from ICU patients, as it is mentioned twice.
11.General Comments:
- English Language: The manuscript would benefit from improved language clarity. I suggest a thorough revision to enhance readability and ensure fluency.
- References: Ensure that the presentation of references aligns with the journal’s guidelines. For example, check the formatting and sequence of citations like [17].
- Abbreviations: Use abbreviations consistently throughout the manuscript. For instance, "REE" is first introduced in line 142, and then reintroduced in line 158. It would be helpful to include a list of abbreviations used in the manuscript for reference.
Comments on the Quality of English Language
Extensive editing of English language required.
Author Response
Comments 1. “Revise the abstract to read as follows: Sepsis is a life-threatening condition characterized by an imbalanced immune response to infection, posing a significant challenge in hospital settings due to its high morbidity and mortality rates. While much attention has been given to patients in the Intensive Care Unit (ICU), uncertainties remain regarding the nutritional management of septic patients in other departments, particularly in Internal Medicine. This narrative review aims to address these gaps by exploring key aspects of nutritional care in sepsis patients admitted to the Internal Medicine department. We examine the pathophysiological mechanisms driving metabolic alterations in sepsis, methods for effective nutritional assessment, and supplementation strategies, including the potential role of specific nutrients. Additionally, we discuss the preventive role of nutrition, with a focus on gut microbiome modulation. By synthesizing the available literature, this review highlights critical areas for future research and provides evidence-based insights to guide nutritional strategies for managing sepsis in Internal Medicine patients”.
Response 1. As suggested, we revised the abstract.
Comments 2. “Lines 32-35: Please rephrase this paragraph as follows: The severity and prognosis of sepsis are influenced by multiple factors, including host characteristics (such as age, race, genetic variability, comorbidities, and medications), as well as infection-related factors, including the type, location, pathogen load, and antimicrobial resistance”.
Response 2. Lines 32-25 - We rephrased this paragraph as suggested by the reviewer.
Comments 3. “Lines 48-49: specify the mortality rates”.
Response 3. Lines 48-49 - As suggested, we specified the mortality rates.
Comments 4. “Line 65: Incorporate a figure legend for figure 1”.
Response 4. Line 65 - As suggested, we incorporated a figure legend for figure 1.
Comments 5. “Line 161: Present a brief comment on the results of the studies discussed to provide a critical interpretation of the findings”.
Response 5. Line 161 - As suggested, we added a brief comment on the results of the studied discussed in this line to provide a clinical interpretation of the findings.
Comments 6. “Line 165: Remove the phrase: (see section 3.2 for more details on refeeding syndrome)”.
Response 6. Line 165 - As suggested, we removed this phrase.
Comments 7. “Line 216: There seems to be missing information regarding the NUTRIREA-2 trial, which may have been accidentally omitted, as the trial is discussed in the following section”.
Response 7. Line 216 – As the authors correctly noted, the NUTRIREA-2 trial is discussed in the following section. The words “The trial NUTRIREA-2” were deleted at this line.
Comments 8. “Line 220: Clarify what is meant by “either testing”. If the authors are referring to tolerance, the following revision may be used (lines 218-221): "According to the 2023 ESPEN guidelines on clinical nutrition in the ICU, EN should be initiated early, provided there are no absolute contraindications. Initially, 20-50% of full nutritional requirements should be administered to either assess tolerance or achieve a trophic effect on the gastrointestinal system. If well tolerated, EN should be progressively increased until the optimal nutritional target is reached”.
Response 8. Line 220 - We were referring to tolerance. So, we changed lines 218-221 as suggested by the reviewer.
Comments 9. “Nutritional Assessment and Treatment: 1. Initially, the authors should present the potential benefits of nutrition in sepsis within the main text. Following this, at the end of the section, a brief subsection should be included that discusses the controversies surrounding nutritional supplements, drawing on evidence from various studies. 2.Incorporate the results of recent studies and meta-analyses to strengthen this section: - On the role of Vitamin C in sepsis (PMID: 36743822). - On the role of Thiamine in sepsis prognosis (PMID: 36259460). 3. General Comment: It would be helpful to include a table summarizing the clinical studies and outcomes related to micronutrient and other disease-specific supplementation. Besides, this table could present additional data on micronutrients not mentioned in the main text, such as Zinc (PMID: 39275159)”.
Response 9. We changed the text following the suggestions of the reviewer, and we added a new table (table 1) summarizing the clinical studies and outcomes related to micronutrient and other disease-specific supplementation, as suggested.
Comments 10. “Lines 423-427: The authors should revise this part to avoid repeating that most data are derived from ICU patients, as it is mentioned twice”.
Response 10. Line 423-427 - We changed the text as suggested.
Comments 11. “General Comments: - English Language: The manuscript would benefit from improved language clarity. I suggest a thorough revision to enhance readability and ensure fluency. - References: Ensure that the presentation of references aligns with the journal’s guidelines. For example, check the formatting and sequence of citations like [17]. - Abbreviations: Use abbreviations consistently throughout the manuscript. For instance, "REE" is first introduced in line 142, and then reintroduced in line 158. It would be helpful to include a list of abbreviations used in the manuscript for reference”.
Response 11. English Language - As suggested, we made an extensive editing of English language. References - As suggested, we presented the references according to the journal’s guidelines. Abbreviations – As suggested, we included a list of abbreviations used in the manuscript.
Round 2
Reviewer 2 Report
Comments and Suggestions for Authors
The manuscript was improved.
1. What is the difference in the pathology of sepsis among patients inside and outside of intensive care units? The difference in strategies derived from their pathologic difference could be more detailed. How is that in patients who transfer from inside to outside of intensive care units?
2. Row 166; what is the ‘adequate’ level in ‘adequate’ protein, energy, and other essential nutrients? The term ‘adequate’ may be ambiguous.
3. The expression ‘OR (without ‘.’) and RR (with ‘,’) could be consistently unified. It could be similarly done for CI.
4. English can be rechecked by native.
Comments on the Quality of English Language
English can be rechecked by native.
Author Response
What is the difference in the pathology of sepsis among patients inside and outside of intensive care units? The difference in strategies derived from their pathologic difference could be more detailed. How is that in patients who transfer from inside to outside of intensive care units?
The pathology of sepsis does not change whether we are in or out of intensive care. Nutritional strategies are clearly a little different and depend on the means available. We have modified the text trying to better emphasize this point.
- Row 166; what is the ‘adequate’ level in ‘adequate’ protein, energy, and other essential nutrients? The term ‘adequate’ may be ambiguous.
We have modified the text as suggested.
- The expression ‘OR (without ‘.’) and RR (with ‘,’) could be consistently unified. It could be similarly done for CI.
We have made the suggested corrections.
- English can be rechecked by native.
English has been rechecked as suggested.
Reviewer 3 Report
Comments and Suggestions for Authors
The manuscript has been significantly improved. It is suitable for publication. A minor observation. The title should be revised as follows:
Nutritional Strategies for the Treatment and Prevention of Sepsis Outside the Intensive Care Unit
Author Response
The title was changed as suggested.